# Uncovering Latent Heterogeneity in Alzheimer's Disease via Clustering of Cortical GNN Explanations

**Ian Liu**[1,2]                                         Ian_y_Liu@brown.edu
[1] *Brown University School of Public Health, Providence, RI, USA*
[2] *Northern Medical Research Institute, Middletown, NY, USA*

**Da Ma**[3]                                            Da.Ma@wfusm.edu
[3] *Wake Forest University School of Medicine, Winston-Salem, NC, USA*

## Abstract

Alzheimer's disease (AD) is clinically and biologically heterogeneous, yet many neuroimaging classifiers reduce this variability to a single disease score. We investigate whether subject-level explanation maps from a cortical graph neural network (GNN) can reveal latent heterogeneity within AD. Using baseline T1 MRI from stable cognitively normal (CN) and AD participants from ADNI, we utilize the best-performing GATv2 from our prior benchmark and create node-feature attribution maps over area, curvature, and thickness. We performed split-safe k-means clustering, with fitting and preprocessing only on training AD subjects. We compared clustering results from raw cortical features, pre-classifier GATv2 embeddings, and GNN explanation maps across $k = 2, ..., 6$ using training-set silhouette score, cluster-size constraints, and anatomical interpretability of held-out assignments. Compared to raw features and graph embeddings, node explanation maps yielded the most interpretable non-degenerate structure, separating AD subjects into two explanation phenotypes with distinct temporal versus more diffuse cortical reliance patterns.

**Keywords:** Alzheimer's disease, graph neural networks, explainability, clustering, cortical surface, subtype discovery

## 1. Introduction

Alzheimer's disease (AD) is heterogeneous in both progression and neuroanatomical expression (Noh et al., 2014; Zhang et al., 2021), yet many neuroimaging studies still frame AD as a homogeneous binary classification problem, emphasizing predictive accuracy while overlooking within-class variation. Cortical morphometry is well suited to studying this heterogeneity because atrophy is spatially distributed across the highly-folded and non-Euclidean geometry of the cerebral cortex. Graph neural networks (GNNs) are particularly suitable because they operate directly on irregular cortical meshes while preserving anatomical locality.

In our earlier benchmark on harmonized cortical morphometry, harmonized GATv2 Brody et al. (2021) emerged as the strongest graph model (test F1 = $0.83 \pm 0.04$, ROC-AUC = $0.93 \pm 0.02$) across stratified nested cross-validation, and its explanation maps consistently emphasized cortical thickness more strongly than curvature or area (Liu and Ma, 2026). Unlike ROI-based GNNs such as functional or structural brain networks, which constrain interpretation to atlas-defined regions and may obscure fine-grained effects near

region boundaries, our mesh-based formulation retains vertex-level detail. The substantial between-subject variability in these explanation maps motivated the present study. Here, we shift the goal from *prediction* to *phenotyping*: rather than asking only whether a GNN can distinguish AD from CN, we ask whether clustering subject-level explanation maps can reveal latent heterogeneity within AD. We hypothesize that the GNN explanation space captures distinct modes of model reliance on cortical morphology and that these modes may correspond to proposed disease subtypes.

## 2. Methods

We analyzed baseline T1 MRI from 1,409 ADNI (Mueller et al., 2005) participants with stable diagnoses across follow-up (54.2% CN, 45.8% AD; mean age 74.4 years). FreeSurfer (Fischl, 2012) produced cortical meshes and three vertex-wise morphometric measures: area, curvature, and thickness. Features from all subjects were surface-registered to *fsaverage* (Fischl et al., 1999), yielding 163,842 vertices per hemisphere.

We applied the harmonized pipeline used in prior work: site effects were reduced with `neuroHarmonize` (Fortin et al., 2018), followed by W-score normalization (Chung et al., 2017) estimated from CN training subjects. Using harmonized GATv2 as the explanation backbone, for each explained subject, GNNExplainer Ying et al. (2019) was applied to the trained graph-level classifier with attribute-level node masking, yielding a node-feature importance tensor of shape $N \times F$, where $F = 3$ corresponds to area, curvature, and thickness. Each subject's explanation tensor was then flattened into explanation space and used as a phenotype representation.

Clustering was performed in a split-safe manner: the representation and clustering model are only fit on training data. We iteratively fitted $k = 2, \ldots, 6$ using Euclidean distance and repeated initialization, then selected $k$ by the maximum training silhouette score subject to a minimum cluster-size constraint. Validation/test AD subjects were standardized with the training scaler and assigned to the nearest training centroid. The workflow merges undersized train clusters to the nearest retained centroid, so selected $k$ and the number of effective reported assignment groups can differ. This same workflow was performed on raw cortical features and pre-classifier graph embeddings.

## 3. Results

Consistent with the earlier benchmark, group-level explanation contrasts emphasized cortical thickness as the dominant discriminative feature. The standard deviation maps suggested greater heterogeneity in AD than CN explanation space, particularly in the thickness channel over temporal cortex (Fig. 1A). Under the split-safe clustering workflow, explanation-map clustering selected k=2 and produced two non-degenerate AD explanation phenotypes with cluster sizes of n=159 and n=51 in the training folds. In our comparison with raw cortical features and pre-classifier GATv2 embeddings, explanation maps yielded the most anatomically interpretable non-degenerate cluster-average patterns, separating subjects along temporal-localized versus more diffuse cortical reliance patterns (Fig. 1B-C). Because these clusters are derived from model attributions rather than direct atro-

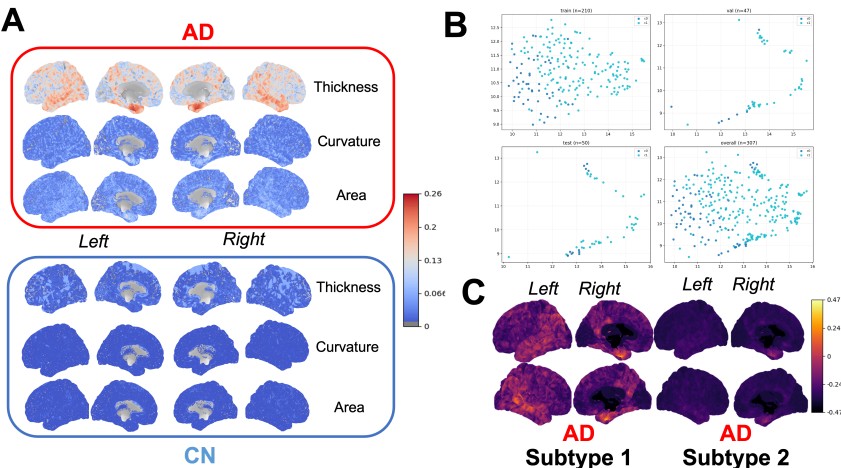

Figure 1: **Results**. **A**: Standard deviation maps of AD and CN explanation values show greater AD variability, particularly in the thickness channel over temporal cortex. **B**: t-SNE (Van der Maaten and Hinton, 2008) visualizations of AD explanation maps colored by k-means clusters across train, val, test, and overall. **C**: Cluster-average cortical thickness explanation maps for two AD explanation phenotypes, showing temporal-localized versus more diffuse model-reliance patterns.

phy measurements, we do not interpret k=2 as evidence that stable AD has exactly two biological subtypes.

## 4. Discussion

We reframe GNN explainability as a representation-learning problem for disease phenotyping. Instead of treating explanation maps only as post hoc visualizations, we use them as subject-level objects that can be clustered to probe latent heterogeneity in AD while retaining anatomical localization. These explanation phenotypes are broadly consistent with prior literature reporting heterogeneous temporal versus more diffuse cortical involvement in AD. Since these clusters are derived from model attributions rather than raw cortical thickness, they should be interpreted as "model-reliance phenotypes," not direct neuroanatomical or neuropathological subtypes. Additionally, our cohort includes only stable CN and stable AD participants; future work should examine whether similar explanation phenotypes emerge in converter populations.

Several caveats remain important. Explanation differences are not the same as raw morphometric differences because they reflect where the model relied more heavily on specific node-feature entries. Likewise, the observed cluster structure should be interpreted as exploratory rather than confirmatory. Even so, these findings suggest that explanation space contains structured variation beyond binary diagnosis and may help bridge prediction and phenotyping in cortical-surface neuroimaging.

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

**Acknowledgments**

Data used in the preparation of this article were obtained from the Alzheimer's Disease Neuroimaging Initiative (ADNI) database. Data collection and sharing for ADNI are funded by the National Institute on Aging under NIH grant U19 AG024904. The grantee organization is the Northern California Institute for Research and Education. ADNI has also received past support from the National Institute of Biomedical Imaging and Bioengineering, the Canadian Institutes of Health Research, and private-sector contributions through the Foundation for the National Institutes of Health. We thank the ADNI participants and investigators whose efforts made this work possible. This work was supported in part by the NIH/NLM CALIBIR program under Award Number R25LM014214. The content is solely the responsibility of the authors and does not necessarily represent the official views of the NIH.

