# OpenReview forum: "Uncovering Latent Heterogeneity in Alzheimer's Disease via Clustering of Cortical GNN Explanations"
_MIDL.io/2026/Short_Papers — MIDL 2026 - Short Papers Poster_

### Official Review · Reviewer_FetR · 2026-04-30
**phenotyping using GNN explanation map**

**Rating:** 4
**Confidence:** 3

**Review:**

see strengths and weaknesses.

**Summary:**

This paper proposes the use of graph neural network derived explanation maps as a representation for clustering and phenotyping Alzheimer’s disease subtypes. Specifically, subject-level explanation maps are generated for vertex-wise cortical features, including surface area, curvature, and thickness. These maps are then used as inputs to a clustering algorithm, with the optimal number of clusters selected based on the silhouette score. The method identifies two clusters that differ in both overall explanation magnitude and spatial distribution patterns.

**Strengths:**

The idea of leveraging explanation maps as features for phenotyping is appealing. By capturing how the GNN assigns importance to different cortical features, the approach provides a potentially interpretable way to identifying disease subtypes. This perspective could stimulate meaningful discussion on model-driven phenotyping in neuroimaging.

**Weaknesses:**

The paper lacks sufficient detail regarding the clustering method. Additionally, while the method identifies two clusters based on the silhouette score, there is limited discussion on whether this number aligns with prior clinical or biological expectations. It remains unclear how sensitive the findings are to the choice of clustering parameters or whether alternative numbers of subtypes could be equally plausible.

**Justification Of Rating:**

While there are some areas that could benefit from additional clarity and detail, the core contribution (using GNN-derived explanation maps as representations for clustering and phenotyping Alzheimer’s disease subtypes) is interesting and potentially impactful.

---

### Decision · Program_Chairs · 2026-05-08

Accept (Poster)